# From 2D to 3D: Development of Monolayer Dopaminergic Neuronal and Midbrain Organoid Cultures for Parkinson’s Disease Modeling and Regenerative Therapy

**DOI:** 10.3390/ijms24032523

**Published:** 2023-01-28

**Authors:** Yee Jie Yeap, Tng J. W. Teddy, Mok Jung Lee, Micaela Goh, Kah Leong Lim

**Affiliations:** 1Lee Kong Chian School of Medicine, Nanyang Technological University, Singapore 308232, Singapore; 2Interdisciplinary Graduate Programme (IGP-Neuroscience), Nanyang Technological University, Singapore 639798, Singapore; 3National Neuroscience Institute, Singapore 308433, Singapore; 4Department of Brain Sciences, Imperial College London, London SW7 2AZ, UK; 5Department of Anatomy, Shanxi Medical University, Taiyuan 030001, China

**Keywords:** Parkinson’s Disease, pluripotent stem cells, dopaminergic neurons, organoids, mesencephalon, midbrain, microfluidic, clinical trial, allogeneic transplantation, regenerative medicine

## Abstract

Parkinson’s Disease (PD) is a prevalent neurodegenerative disorder that is characterized pathologically by the loss of A9-specific dopaminergic (DA) neurons in the substantia nigra pars compacta (SNpc) of the midbrain. Despite intensive research, the etiology of PD is currently unresolved, and the disease remains incurable. This, in part, is due to the lack of an experimental disease model that could faithfully recapitulate the features of human PD. However, the recent advent of induced pluripotent stem cell (iPSC) technology has allowed PD models to be created from patient-derived cells. Indeed, DA neurons from PD patients are now routinely established in many laboratories as monolayers as well as 3D organoid cultures that serve as useful toolboxes for understanding the mechanism underlying PD and also for drug discovery. At the same time, the iPSC technology also provides unprecedented opportunity for autologous cell-based therapy for the PD patient to be performed using the patient’s own cells as starting materials. In this review, we provide an update on the molecular processes underpinning the development and differentiation of human pluripotent stem cells (PSCs) into midbrain DA neurons in both 2D and 3D cultures, as well as the latest advancements in using these cells for drug discovery and regenerative medicine. For the novice entering the field, the cornucopia of differentiation protocols reported for the generation of midbrain DA neurons may seem daunting. Here, we have distilled the essence of the different approaches and summarized the main factors driving DA neuronal differentiation, with the view to provide a useful guide to newcomers who are interested in developing iPSC-based models of PD.

## 1. Introduction

Parkinson’s Disease (PD) is a major neurodegenerative disorder that affects nearly 10 million people globally and one that recorded the fastest rise in prevalence in recent years [1]. The main histopathological hallmark of PD is the accumulation of α-Synuclein (α-Syn)-positive Lewy bodies in the substantia nigra pars compacta (SNpc) of the midbrain, where dopaminergic (DA) neurons that succumbed to the disease reside [2]. There are three unique subpopulations of DA neurons in the ventral mesencephalon (midbrain) region, i.e., the A8 DA neurons in the retrorubral field, the A9 DA neurons in the SNpc, and the A10 DA neurons in the ventral tegmental area (VTA) [3]. A8 and A10 DA neurons are part of the mesolimbic pathway that regulates emotion and motivation, and they project into the ventral striatum, septum, nucleus accumbens, and prefrontal cortex [4], whereas the A9 DA neurons in the SNpc innervate the putamen of the basal ganglia to form the nigrostriatal pathway that coordinates voluntary motor movements [4,5]. In PD, progressive neurodegeneration of A9 DA neurons occurs [6], whereas those in the VTA of the SN are largely unaffected [7,8]. Consequently, PD patients exhibit classical motor symptoms such as bradykinesia, rigidity, and resting tremors that form the basis for clinical diagnosis [9].

Given the vital role of DA neurons in PD, there is a great interest in generating human DA neurons in vitro to use them as a disease model to better understand the pathogenesis of PD and to facilitate drug discovery efforts, as well as a cellular resource to replace the lost DA neurons in PD patients. This has been fueled in large part by the advent of the induced pluripotent stem cell (iPSC) technology that has allowed PD models to be created from human control- and patient-derived cells. Further, the iPSC technology has also enabled the establishment of 3D brain organoids for the study of neurodegenerative diseases. These brain organoids can allow for the study of spatial interactions between various cells on a dish without requiring the invasive access to tissue from brain biopsies. In addition, the diversity of cells that are generated in these organoids can help to uncover complicated mechanisms underlying dysfunction in cell–cell crosstalk and omics changes in neurodegenerative diseases such as PD [10,11]. There is thus much excitement as we enter into a new era of disease modeling for PD using human-derived cells. However, to navigate the plethora of reported protocols that are related to DA neuronal generation from pluripotent stem cells (PSCs) can be quite daunting and confusing at times, especially for newcomers to the field. Indeed, a comprehensive review by Marton et al. presented a summary of more than 70 available protocols for deriving DA neurons in a 13-year timeframe from 2004 to 2017, of which 65% were not used by other studies [12]. Despite the high redundancy, new protocols are still being developed with the motivation of further increasing the yield of DA neurons derived from iPSCs [13,14]. One of the major aims of this review is therefore to consolidate the various in vitro DA differentiation protocols in terms of how closely they mimic the molecular mechanisms driving DA neuronal development in vivo and summarize the key molecules involved, with the view to provide a useful guide to newcomers who are interested in developing iPSC-based models of PD. At the same time, we also provide an update on the latest advancements in using these cells for drug discovery and regenerative medicine.

## 2. The Three Stages of Midbrain DA (mDA) Neuronal Development

Regardless of 2D or 3D modeling, in order to generate DA neurons on a dish, various research groups have taken extensive efforts to understand the signaling pathways underlying the development of the ventral midbrain starting from the neurulation process. Neurulation signals the beginning of neurodevelopment in an embryo, where the lateral margins of the neural plate folds inwards to establish the neural tube [15]. The ventral midline of the neural tube differentiates to lay the floor plate while differentiation of the dorsal midline of the neural tube initiates the formation of the roof plate. The roof plate and floor plate alongside the surrounding notochord and somite serve as important signaling centers for neurodevelopment [16]. In particular, the derivation of DA neuronal progenitors originates from cells at the floor plate of the ventral midbrain [17] that is driven by several signaling molecules such as sonic hedgehog (Shh), retinoic acid (RA), noggin from the floor plate, and fibroblast growth factor (FGF) from the nearby presomitic mesoderm [16]. These factors, together with the plethora of signals, collectively drive the neurodevelopmental process of DA neurons via 3 main stages: 1. specification of midbrain floorplate; 2. specification of midbrain DA (mDA) neurons/mDA neurogenesis (and suppression of alternative fates); and 3. mDA maturation and survival [18,19] (Figure 1).

### 2.1. Stage 1—Specification of Ventral Midbrain Fate Driven by Otx2-Wnt1, Gbx2-Pax2-FGF8, and Shh-FoxA2

Formation of the isthmic organizer (IsO) early in the neurodevelopment process is an essential event in specifying the fate of the midbrain [18]. The IsO is an important signaling center that defines the midbrain-hindbrain boundary (MHB). As early as 7.5 days of embryonic age, transcription factor Otx2 (orthodenticle homolog 2) expression in the midbrain and Gbx2 (gastrulation brain homeobox 2) expression in the hindbrain trigger the pathway that establishes the IsO and the MHB [20]. Otx2 regulates Wnt1 (wingless-int1) expression in the midbrain [21], whereas Gbx2 regulates FGF8 expression through Pax2 (paired homeobox 2) in the hindbrain [22]. The midbrain-hindbrain spatial specific expression is maintained by a mutually dependent feedback system between Wnt1 and FGF8 [23]. Interestingly, FGF8 alone is necessary and sufficient for the formation of the IsO [24], which in turn secretes more FGF8 to establish a FGF8 gradient [18]. The regulation of FGF8 activity at the IsO is apparently regulated by Wnt [25]. Cells at the higher end of the FGF8 concentration gradient are driven to a hindbrain fate whereas cells at the lower end of the gradient assume a midbrain fate [26]. The second major signaling event in midbrain patterning involves Shh-FOXA2. Shh signaling occurs when Shh binds to and inactivates the transmembrane protein Patched, thus freeing and thereby promoting the accumulation of Smoothened on the cell surface (as Patched normally prevents Smoothened from being activated) [16,27]. This activates Gli1 and Gli2 translocation into the nucleus that consequently upregulates the expression of ventrally expressed homeodomain transcription factors [18,27]. Notably, FOXA2 is a key transcription factor upregulated by robust Shh signaling [28]. FOXA2 is proposed to repress the activity of Hedgehog family of proteins that drives the dorsal fate. Thus, in the absence of FOXA2, the midbrain adopts a dorsal fate instead of a ventral fate [29]. Hence, FOXA2 is critical for the development the ventral midbrain fate from which mDA neurons develop.

### 2.2. Stage 2—Specification of mDA Neurogenesis and Suppression of Alternative Fates

Many of the transcription factors that drive the ventral midbrain floor plate development (i.e., Otx2, Wnt1, FGF8, Shh, and FOXA2) also play integral roles in mDA neuron specification. For instance, Wnt1, FGF8, and Shh are all required for the ectopic induction of mDA neurons [30]. Loss of Wnt1 will result in the loss of proliferating mDA progenitors, which highlights the crucial role Wnt1 plays in the specification, differentiation, and maintenance of mDA progenitors [30]. Wnt1 at the midbrain floor plate ensures the mDA fate via activating the Otx2-Wnt1-LMX1A/Msx1 pathway [18]. Wnt1 also maintains Otx2 expression that is required to repress Nkx2.2, a product of Shh from the floorplate that promotes serotonergic fate [30]. Failure to repress Nkx2.2 in the midbrain will lead to the generation of serotonergic, rather than DA neurons [31]. In addition to suppressing alternative neuronal fates, Otx2 and its downstream factor LMX1A function as important determinants of mDA fate [32,33]. Indeed, ectopic expression of LMX1A is sufficient to induce mDA neurons in chick embryos [34]. LMX1A first induces the expression of Msx1, and together they activate the proneural factor Ngn2 (neurogenin 2) and the basic-helix-loop-helix family, Mash1 (mouse achaete-schute homolog 1) that further drive mDA neurogenesis [34,35]. Ngn2 activation results in Sox2-positive progenitors that mature into Nurr1-positive neurons and eventually differentiate into tyrosine hydroxlase-positive (TH+) mDA neurons [36]. At the same time, LMX1A also coordinates the specification of mDA neurons by cooperating with FOXA2 [33]. On top of determining a ventral midbrain fate, FOXA2 is key to DA neuron specification. FOXA2-null embryos do not survive past the embryonic age of 10.5 days, whereas FOXA2 overexpression results in a four times increase in the number of TH+ cells [17]. FOXA2 induces the differentiation of both the red nucleus and DA neurons in the SNpc, and LMX1A selects for the DA fate by repressing the red nucleus Sim1-Lhx1-Ngn1 pathway [33]. In addition, LMX1A assists FOXA2 in developing the floor plate by repressing Nkx6.1 that inhibits floor plate differentiation [33]. FOXA2 can also drive DA fate by collaborating with Otx2 in inhibiting Nkx2.2 that otherwise promotes the development of serotonergic neurons [37]. Finally, as mentioned above, mDA neurogenesis is further driven by proneural factors Mash1 and Ngn2 [38,39], with Mash1 capable of partially compensating for the loss of Ngn2 [36]. Supporting this, a reported study [36] revealed that the loss of Ngn2 decreased the mDA neuronal population by 50% in wildtype mice, with a further decrease in mDA neuron numbers upon the deletion of Mash1. As can be appreciated from Figure 1, Ngn2/Mash1 is a convergent point for the Otx2-Wnt1-LMX1A/Msx1 pathway [39]. FOXA2 also converges on Ngn2/Mash1 via the Msx1-LMX1A pathway (Figure 1). In parallel, FOXA2 can directly regulate Ferd3l, which represses Hes1 that suppresses proneural genes to promote proneural development [32]. Finally, although Ngn2 is critical for mDA neurogenesis, it is unable to specify the terminal differentiation into TH+ mDA neurons [40], which requires other factors as discussed below.

### 2.3. Stage 3—mDA Maturation Requires Nurr1, Pitx3, En1 and Survival Requires Gdnf, Bdnf, and Tgfβ

Terminal differentiation of neural cells into TH+ mDA neurons requires Nurr1, En1, and Pitx3 [41,42]. The central pathways described above involving the Wnt-LMX1A pathway regulates the expression of Nurr1 and Pitx3 [43], while that of FOXA2-related components are required for the expression of Nurr1, En1 (Engrailed 1), Ddc (dopa decarboxylase), and TH in mDA neurons [44]. Nurr1 is essential for TH expression [45] and mDA neuronal survival [46]. In the absence of Nurr1, Pitx3-positive neuroblasts failed to survive [46]. Aside from TH, Nurr1 also regulates the expression of characteristic mDA neuronal markers including neurotransmitter vesicle membrane packaging transporter VMAT2, cell membrane dopamine reuptake transporter DAT, and dopamine decarboxylase DDC [18]. Another molecule important for DA neuronal differentiation is En1, a homeodomain transcription factor that is ubiquitously expressed in the rostral midbrain [47]. In En1 mutants, both SNpc and VTA were devoid of DA neurons [48], a phenomenon that amply illustrates its importance to mDA biogenesis. En1 complements Otx2 in the later stages of the formation and maintenance of the midbrain-hindbrain boundary (MHB). This apparently takes place via the positive regulation of Wnt1 by En1 by way of destabilizing β-catenin [49]. En1 is key to the rostral-caudal landscaping of the midbrain that gives rise to the SNpc and VTA, respectively [41]. When En1 is lost, expression for Cck (cholecystokinin) and Aldh1a1 alongside its resultant retinoic acid (RA) production is downregulated or lost, resulting in a lack of topographical identity important for rostro-caudal patterning [41]. Finally, for A9-specific DA neurons to develop, the participation of the homeobox transcription factor Pitx3 is critical. Pitx3 is essential for A9 mDA neuronal survival in the SNpc, as its deletion does not appear to affect VTA A10 mDA neurons [50]. Pitx3 upregulates Aldh1a1 (aldehyde dehydrogenase-1a1) that in turn upregulates TH and Drd2 (dopamine receptor 2) via a RA-dependent mechanism. Additionally, Pitx3 can also upregulate key mDA markers Aldh1a1, VMAT2, and DAT, and downregulate Cck and En1 in an RA-independent manner pathway [51]. Upregulation of Aldh1a1 that enhances RA production is important in rostral mDA patterning and regional specification of SNpc [52]. The main function of Pitx3 is to repress En1 and its resultant Cck expression to switch the caudal VTA fate into a rostral SNpc fate [47]. In Pitx3-knockout embryos, En1 is upregulated and mDA neurons were driven to caudal fate. As a consequence, the number of TH+ neurons in the rostral midbrain were reduced, which could be rescued by RA [53]. Thus, a fine interplay between the actions of Pitx3 and En1 is needed for the proper development of SNpc and VTA DA neurons. Another key molecule is Reelin, which promotes the fast lateral-tangential migration of mDA neurons into the SNpc [54]. Interestingly, only Sox6+ mDA neurons migrate to the lateral SNpc, while Otx2+ neurons remain at the VTA [55]. The exact mechanism behind how Otx2 and Sox6 are involved in mDA neuron migration is still unclear, but it will be interesting to further investigate how these factors interact with Reelin.

To summarize, the final rostro-lateral location of the ventral midbrain where A9 mDA neurons reside [56] is due to the sophisticated signaling of a group of transcription factors highlighted above. FGF8, FOXA2, and the repression of En1 by Pitx3, along with Aldh1a1 expression determine the midbrain, ventral, and rostral fate, respectively. After terminal differentiation of mDA neurons by key regulators Nurr1, Pitx3, and En1, the survival and maintenance of mDA neurons are dependent on separate families of neurotrophic factors such as TGFβ3 (transforming growth factor) [57], GDNF (glial cell-line-derived neurotrophic factor) [58], and BDNF (brain-derived neurotrophic factor) [59]. Non-functional TGFβ3 in a rodent model was shown to have a 20% decrease in TH+ DA neurons in the SNpc [60]. In a similar fashion, deletion of GDNF and BDNF were shown to have a pronounced 60–70% and 23% reduction in TH+ DA neurons, respectively [61,62]. Notably, these three key neurotrophic factors work synergistically together to exert their effects [63]. Interestingly, BDNF appears to be additionally regulated by GDNF via a GDNF-Pitx3-BDNF trophic loop [64], which might explain the greater number of TH+ DA neuronal loss in GDNF-knockout versus a BDNF knockout animal. Related to this, as Nurr1 is known to regulate expression of the GDNF receptor Ret and also the GDNF Family receptor alpha [65,66], it is not surprising to note the important role Nurr1 plays in DA neuronal survival.

### 2.4. A9 SNpc DA Neurons and Their Selective Vulnerability to Degeneration in PD

The conventional classification of mDA neurons into the A8, A9, and A10 subgroups according to their locations and projection areas does not fully explain their differential vulnerability in PD, as mDA neuron degeneration is mediated by a complex interplay of intrinsic and extrinsic factors [67]. It is, however, well-established that the A9 DA neurons in the SNpc display greater susceptibility to degeneration in PD than the A10 DA neurons in the VTA, mainly due to physiological and metabolic heterogeneities between the two mDA subpopulations [68,69]. In A9 DA neurons, the constant large intracellular Ca^2+^ oscillations related to their autonomous pacemaker activity and their poor intrinsic Ca^2+^-buffering capacity, owing to the relatively low expression levels of Ca^2+^-binding proteins calbindin-D28k and calretinin, are thought to increase their propensity to degeneration [70,71]. The elevated cytosolic Ca^2+^ levels may trigger Ca^2+^-dependent activation of calpain and caspase proteases involved in the necrotic and apoptotic cell death pathways [72]. Moreover, sustained elevated levels of cytosolic Ca^2+^ are known to provoke a mitochondrial Ca^2+^ overload and a concomitant increase in mitochondrial reactive oxygen species (ROS) production, which in turn induces mitochondrial dysfunction implicated in the degeneration of mDA neurons and PD pathogenesis [73,74]. In contrast, the A10 DA neurons in the VTA exhibit not only higher levels of calbindin-D28k, but also lower L-type Cav1.3 Ca^2+^ channel density, which overall results in reducing the basal level of oxidative stress in these DA neurons [70,75]. Another reason for the selective neuronal susceptibility of A9 neurons in PD concerns the enhanced state of oxidative stress linked to DA metabolism. Active DA auto-oxidation occurs in A9 neurons (that correlates with their high content in neuromelanin, i.e., a by-product of DA oxidative metabolism), which, together with enzymatic degradation of DA to toxic intermediate metabolites, contribute to an increased production of ROS implicated in neuronal degeneration [76]. Furthermore, DA auto-oxidation generates highly reactive *o*-quinone species including DA *o*-quinone, aminochrome, and 5,6-indolequinone, which are capable of modifying and damaging intracellular DNA and macromolecules [77,78]. Considering the significance of DA metabolism in neuronal redox-homeostasis and cell viability, it may be relevant to investigate the differences in DA homeostasis among the mDA subpopulations to further characterize their distinct vulnerability profile. DAT and VMAT2 in particular have been implicated in regulating the spatial-temporal dynamics of DA transmission [79]. While the rapid uptake of DA from the extracellular space into the presynaptic neuron by the DAT terminates DA signaling, this also translates to an increased intracellular DA load, which may result in oxidative stress and neurotoxicity [80]. Indeed, an animal study by Masoud et al. demonstrated that DAT overexpression led to an increase in DA metabolism and oxidative stress markers associated with neuronal loss in the SNpc [81]. Post-mortem brain neurochemical data from 17 neuropathologically confirmed cases of end-stage idiopathic PD have shown that VMAT2 functions counteract the cytosolic DA accumulation via sequestration into vesicles, hence preventing the subsequent conversion of DA to its neurotoxic species [82,83]. A loss-of-function study by Bucher et al. demonstrated that dysregulated DA metabolism upon the build-up of intracellular DA following suppressed VMAT2 expression aggravated nigrostriatal dopaminergic neurodegeneration, which was rescued by reintroducing exogenous VMAT2 [84]. Apart from function-related susceptibility, a recent transcriptomic analysis compared between ventral Sox6+ and dorsal Sox6− DA neurons performed by Pereira Luppi et al. suggested that Sox6 expression, which defines A9 neurons, confers specific vulnerability for PD [85]. Sox6+ DA neurons are apparently enriched with the expression of PD risk loci genes, namely Kcns3, Satb1, FGF20, Kcmp3, Syt17, and Rit2, while Sox6− DA neurons display enriched expression of neuroprotective genes Vglut2 and Calb1 [85]. Given the exquisite vulnerability of A9 DA neurons in PD, the motivation by many is to generate this specific subtype of neurons in vitro via the iPSC approach as an appropriate disease model for PD.

### 2.5. Mimicking DA Neuronal Development In Vitro—A Quick Survey of Current Protocols

As discussed above, a plethora of signaling molecules work intricately together in vivo to programme neural progenitors to adopt the mDA neuronal fate, which researchers have attempted to imitate in vitro using iPSCs. Recently, Marton et al. surveyed and summarized a group of 2D DA neuronal differentiation protocols and reported a total of 27 different small molecules that were used to induce mDA neurons across all reported protocols [12]. These include cellular factors such as Shh, TGFβ3, BDNF, GDNF, cAMP, FGF8, FGF2, Noggin, SAG (sensitive to apoptosis gene), Dorsomorphin, IL1B, Fgf20, EGF, Heparin, Laminin, Wnt1, RA, LIF, SDF1a, sFRP1, and VEGFD, and synthetic compounds such as SB431542, Purmorphamine (agonist of smoothened), CHIR99021 (GSK3 inhibitor), Ascorbic Acid, DAPT, and LDN193189 (inhibitor of BMP signaling). Among these, the most commonly used small molecules are Shh, BDNF, GDNF, Ascorbic acid, cAMP, FGF8, and FGF2 that were reported in more than 46% of the reported protocols for the generation of DA neurons. As discussed above, these are molecules that are involved in the formation of key structures during mDA differentiation in vivo. To reiterate, FGF8 is involved in the formation of the midbrain region while Shh induces the Shh-FOXA2 pathway which enhances the expression of LMX1A [86] to activate pro-neural factor Ngn2 and downstream Nurr1 to drive the TH+ mDA fate [26,33,36,41], whereas BDNF and GDNF are important neurotrophic factors that help to maintain mDA neuronal survival after terminal differentiation [58,59]. Interestingly, the better performing mDA differentiation protocols reported by Kriks et al., Niclis et al., and Gantner et al., in terms of final TH+ neuronal yield (75%, 83%, 80%, respectively) all used a highly similar concoction of signaling molecules, with Kriks et al. using an extra FGF8 signaling molecule step [14,87,88]. In all three protocols, the stepwise exposure of hPSCs to LDN193189 and SB431542, followed by Shh and Purmorphamine and then CHIR99021 were used for mDA induction until about day 11 to 13 of differentiation before switching to the maturation media containing BDNF, GDNF, DAPT, Ascorbic acid, TGFβ3, and cAMP [12,14,87,88] (Figure 2). LDN193189 and SB431542 are inhibitors of TGFβ and BMP signaling and thus inhibit SMAD signaling [89,90]. Inhibition of SMAD signaling initiates neural development in early embryos [91] and is important for neural conversion from human PSCs. The timed activation of Wnt signaling under SMAD inhibition to induce neural crest lineage is crucial. To mimic that, Wnt/β-catenin signaling was activated by adding a GSK3 inhibitor, CHIR99021 [92], a few days after SMAD inhibitors LDN193189 and SB431542 were added. Protocols that used this approach generally had higher percentage of TH+ cells over total cells [12]. Although nanomolar concentrations of CHIR99021 are sufficient to inhibit GSK3, micromolar concentrations are more commonly used, which however carries the risk of potential off-target effects [92]. It will be interesting to examine if lower concentration of CHIR99021 might be able to increase the yield of mDA neurons. Nonetheless, in the presence of GSK3 inhibitors such as CHIR99021, mDA differentiation can occur without adding Fgf8 [93], a key molecule for ventral midbrain fate. This is because transcription of Fgf8 is positive regulated by Wnt/β-catenin signaling that is activated in the presence of CHIR99021 [94]. Clearly, the more successful protocols do imitate closely the key in vivo induction signaling pathways involving Wnt, Fgf8, and Shh. However, one key molecule that is observed in vivo but not used in vitro is the A9 specific transcription factor, Sox6, which would be interesting to explore in future work.

### 2.6. From 2D to 3D—Generating Midbrain Organoid Models of PD

iPSC-derived 2D neuronal progenitors and neurons have contributed significantly to our understanding of PD and provided optimistic prospects with regard to their utility in personalized cell replacement therapy (discussed in later sections). However, 2D neuronal cultures are unable to recapitulate the human brain’s complex physiology and organization. Although several groups have introduced the co-culturing of DA neurons with glial cell types such as astrocytes [96,97,98,99], the monolayer interactions between neuronal cells and non-neuronal cells do not adequately reflect those happening in a spatially organized 3D brain architectural environment. Moreover, it is challenging to maintain 2D cells in long-term cultures due to their propensity for detachment, which renders them not well suited for modeling chronically progressive neurodegenerative diseases. In response to the inadequacies of the 2D monolayer cell cultures, 3D brain organoids were developed [100]. Brain organoids are aggregates of various neuronal and non-neuronal subtypes of cells that develop organized and distinct brain regions. The cytoarchitectural, epigenomic [101], and proteomic [102] semblance between human brain organoids and human fetal brains, as well as their ability to be maintained in culture for indefinite periods of time, provide leverage for the utility of organoids to model neurological and neurodevelopmental disorders. In fact, neurons in brain organoids can generate coordinated electrical oscillations that resemble brain wave patterns observed in newborns [103]. Further, through the use of different combinations of patterning factors, one could generate a myriad of region-specific brain organoids that mirrors the forebrain [104], brainstem [105], cerebellum [106], choroid plexus [107], choroid plexus [107], pituitary [108], thalamus [109], hippocampus [110,111], and midbrain [95,112]. Relevant to this review, we shall focus our discussion on midbrain organoids (MOs) as a 3D model for PD.

In general, both 2D- and 3D-related protocols for generating mDA neurons require dual-SMAD inhibition, followed by exposure to SHH, GSK3 inhibitor CHIR99021, and FGF8 [95,113] (Figure 2). A list of the different small molecules used in 2D [14,86,87,88,96,114,115,116,117,118,119,120,121,122,123,124,125,126,127,128,129,130,131,132,133,134,135,136,137,138,139,140,141,142,143,144,145,146,147,148,149,150,151,152,153,154,155,156,157,158,159,160,161,162,163,164,165,166,167,168,169,170,171,172,173,174,175,176,177,178,179,180,181,182,183,184,185,186,187,188] and 3D-related protocols [4,11,95,112,189,190,191,192,193,194,195,196,197,198,199,200,201,202,203,204,205,206,207,208,209,210,211,212,213,214,215,216] is summarized in Appendix A, respectively, with their respective functions outlined in Appendix A [16,24,87,89,127,136,217,218,219,220,221,222,223,224,225,226,227,228,229,230,231,232,233,234]. However, notwithstanding the slight nuances in the types of small molecules used and the timing the organoids are exposed to with these molecules, one of the main differences between 2D and 3D dopaminergic neuronal generation is that in the latter, stable PSCs or NPCs are dissociated into single cells, seeded in defined quantities in ultra-low attachment 96 wells, and exposed to mediums supplemented with rock inhibitor for a day or two to reduce anoikis (i.e., apoptosis due to loss of cellular attachment to matrix). If successful, the embryoid bodies (EBs) formed will self-organize into spherical structures that will subsequently be exposed to factors that influence floor plate induction and midbrain patterning in a manner similar to those utilized in 2D dopaminergic neuronal generation. Unlike 2D cultures, EBs may be embedded in hydrogels that mimic the extracellular matrix membrane (ECM) such as Matrigel [95,189,191,194,215] or geltrex [197] prior to long term maturation on an orbital shaker or in a bioreactor. Some protocols may skip the embedding step to ensure reproducibility and homogeneity by minimizing manual intervention [235], while others generate EBs in AggreWell^TM^ plates and allow cells to generate their own ECM [112]. Additionally, 2D and 3D cultures share similar markers of differentiation at defined time points, although 3D cultures invariably have a richer cell diversity as compared to 2D cultures, especially in human MOs (hMOs) that are past DIV60. In hMOs, there is a clear temporal sequence in the appearance of various cell types, as corroborated by transcriptomic data, starting with the appearance of markers of floor plate-like cells and early DA progenitors (FOXA2, OTX2, LMX1A, CORIN, MSX1), followed by markers of midbrain-fate neurons (NURR1, MASH1, EN1, TH, PBX1, DDC), and finally the appearance of mature DA neuronal markers typically at post-DIV60 of differentiation (GIRK2, TH) that can be of A9 (GIRK2) or A10 (CALB1) subtype. This is accompanied subsequently by stepwise development of vascular leptomeningeal cells (VLMCs), astrocytes, and oligodendrocyte progenitors [113]. Single cell transcriptomic analysis has revealed a high degree of similarity between the cellular diversity in hMOs with the cell types present in an adult human midbrain, which support the viability of using hMOs in disease modeling and therapeutics discovery for PD. Furthermore, hMOs also contain neurons that can produce neuromelanin and secrete dopamine, similar to physiological conditions [95,113].

hMOs can be derived from a variety of PSCs [215] including embryonic stem cells (ESCs) [112,194] and iPSCs [95], or from neural progenitor cells (NPCs) [189,191,197,235]. The advantages and disadvantages of using ESCs and iPSCs for modeling have been reviewed by Halevy and Urbach [236]. In essence, patient-derived iPSCs may be preferred over ESCs for disease modeling because iPSCs are able to more faithfully recapitulate the pathological features as they are patient-specific. Moreover, iPSCs are free from the ethical implications associated with ESC cultures. On the other hand, there are a few features of iPSCs that might confound their use for disease modeling. These include, but are not limited to, the oncogenic potential of factors used for reprogramming [237], retention of epigenetic memory attributed to incomplete reprogramming [238], and somatic coding mutations [239], as well as genetic variations such as genome instability, copy number variations, and single nucleotide variations [240]. Despite these differences, it seems that to date, hESCs and hiPSCs are both widely used for disease modeling (Appendix A). A modification of these is the use of NPCs for hMO generation. NPCs have already been patterned towards the midbrain fate, which helps to reduce the risk of variation across organoids and ensure reproducibility among different batches. This is because of the oft-significant variation among the organoids established even when they are derived from the same PSC line. Another consideration in reducing heterogeneity between organoids lies with the choice of factors used in each differentiation step. For instance, Kwak and colleagues found that in hMOs exposed to dorsomorphin and A83-01 during the dual SMAD inhibition step to trigger floor plate induction, more than 85.7% of MAP2-positive neurons were TH+ positive, as compared to those exposed to the more commonly used pairs of Noggin and SB431542, or LDN193189 and SB431542 [194]. In contrast, the latter two pairs of differentiation factors are associated with a more heterogenous distribution of TH+ mDA neurons and a higher activation of markers related to the cerebral cortex and non-neuronal genes [194]. More recently, Renner and colleagues [198,235] reported that the removal of the matrigel embedding step in their protocol enables the generation of homogenous hMOs in terms of size, shape, protein expression, and functionality. However, they also reported that their NPC-derived MOs lacked the presence of complex structures as compared to hiPSC-derived hMOs, citing the committed and thus restricted fate of the NPCs as an explanation [198]. Clearly, the ability to generate homogenous organoids remains a challenge that warrants further optimization.

### 2.7. Using MOs in PD Disease Modeling and Drug Discovery

The ability of MOs to recapitulate key pathological features of PD with considerable fidelity has been widely heralded as a breakthrough that has ushered in a new era of experimental PD modeling. Several groups including ours have reported the characterization of hMOs generated from idiopathic PD patients as well as those harboring PD-related mutations or variants including SNCA, LRRK2, PRKN/Parkin, PINK1, DNAJC6, and GBA1 that recapitulate key features of the disease (Appendix A). For example, hMOs generated from idiopathic PD patients exhibited a decrease in growth after DIV49, that is accompanied by a 5-fold reduction in TH levels associated with DA neurodegeneration. Moreover, these PD MOs exhibit enhanced levels of PTX3 (pentraxin-related protein), which is consistent with reports citing a positive correlation between increased PTX3 levels and degree of PD severity in PD [192]. Alongside this, we have shown that hMOs carrying GBA1^−/−^ or SNCA duplication display an increased number of TH+ mDA neurons harboring α-Syn aggregates and concomitantly, a higher expression of apoptotic markers that is related to a reduction of mDA neurons over time. Importantly, we further demonstrated that hMOs containing both GBA1^−/−^ and SNCA mutations are able to recapitulate Lewy body-like inclusions in vitro, which mimics the histopathological hallmark of PD [204]. In a related study, Mohamed and colleagues reported that the levels of oligomeric and phosphorylated α-Syn aggregates increased with time in a hMO model harboring SNCA gene triplication, although the authors did not observe the formation of Lewy body-like structures in their system [201]. This is consistent with our observation above, where Lewy body-like inclusions are rarely seen in hMOs derived from GBA1^−/−^ or SNCA overexpressing PSCs but appear more frequently in GBA1^−/−^/SNCA double mutant organoids [204]. Besides SNCA, LRRK2-related PD hMO models have also been generated. In one study, hMOs with LRRK2-G2019S mutations recapitulate disease phenotypes that includes a reduction in complexity as well as the number of mDA neurons. Interestingly, the same study revealed a significant increase of FOXA2-positive progenitor cells in the patient-specific organoids, which the authors proposed to be a compensatory response to counteract defective specification of mDA neurons induced by the LRRK2-G2019S mutation [191]. Whether this compensatory mechanism observed in hMO culture may happen in real life in as interesting question to think about, especially in view of recent findings demonstrating the relevance of brain organoids grown on a dish in revealing the trajectories of development [241]. Supporting a role for LRRK2 in neurodevelopment, Zagare et al. recently presented evidence that LRRK2 p.G2019S mutant organoids exhibit reduced cellular variability as a result of untimely and incomplete differentiation [11]. The authors performed transcriptomic analysis of these organoids and proposed that DNAJC12, alongside APP, GATA3, and PTN, are likely candidates that contribute to the observed transcriptome changes in LRRK2 mutant organoids that occur during early neurodevelopment [11]. Interestingly, several of the DNAJC family of proteins besides DNAJC12 are also linked to PD. These include DNAJC5 (CSPα) [242] and DNAJC6 (Auxilin) [243]. Notably, CRISPR-edited hMO models of familial juvenile parkinsonism carrying homozygous loss-of-function mutations of DNAJC6, unlike Dnajc6-knockout mice, showed early ventral midbrain patterning defects, downregulation of key mDA neurodevelopmental genes, loss of DNAJC6-mediated endocytosis resulting in aberrant WNT-LMX1A signaling, and increased α-Syn pathology alongside mitochondrial and autolysosomal defects [206]. Similar to DNAJC6 mutant organoids, hMOs carrying another autosomal recessive PD gene, i.e., PRKN, also show abnormalities including being smaller in size compared to those derived from age- and sex-matched control counterparts as well as features indicative of DA neurodegeneration [193]. Further, hMOs with PRKN mutations also exhibit a reduction in GFAP and S100B-positive astrocytes that is similar to the astrocytic alterations observed in postmortem brains of individuals with PRKN mutations [193], suggesting the involvement of a non-autonomous DA neuronal cell death mechanism related to astrocytic dysfunction in the brains of PRKN-related patients. Related to this finding, Dong et al. recently measured the protein turnover rate in control and Parkin-deficient hMOs and found that among the 773 proteins analyzed, only about 6% of proteins were significantly different from the control and mutant organoids [208]. These include vimentin, an intermediate filament and astrocyte marker found to be altered in the earlier-mentioned study [193].

Aside from disease modeling, PD MOs clearly represent a useful platform for testing the neurotoxic effects of compounds. For example, MOs exposed to the gut metabolite trimethylamine N-oxide—that was found elevated in the midbrains of PD patients—trigger neurodegeneration phenotypes including the loss of DA neurons, accumulation of neuromelanin, aberrant BDNF signaling, and astrocyte activation [210]. Indeed, hMOs are used as viable models for such toxicity testing [194,196,203,212], and even in machine learning-assisted neurotoxicity prediction [195]. Alongside this, MOs also represent an excellent resource to evaluate the therapeutic effects of drugs. Using MOs derived from a young-onset PD (YOPD) patient who exhibits defective vesicular catecholamine storage related to α-Syn upregulation, Zhu et al. showed that treatment of these patient-derived organoids with amantadine and phorbol 12-myristate 13-acetate ameliorate the dysfunction of vesicular storage in DA neurons derived from the YOPD patient [216]. It is noteworthy to mention that amantadine is currently being used a clinical drug for PD to enhance dopamine release. Interestingly, 2-Hydroxypropyl-B-Cyclodextrin, a solubilizing agent of lipophilic compounds that is currently used on a Phase 2b/3 clinical trial for Niemann Pick disease (an autosomal recessive lysosomal storage disorder), was found to mitigate the disease-associated phenotypes of PD hMOs with biallelic pathogenic PINK1 variants, but in this case, its action is apparently via enhancing autophagy and mitophagy capacity in PINK1-deficient neurons [209]. Finally, to accelerate the drug discovery process, Ha et al. developed a method to generate simplified brain organoids (simBOs) in just two weeks that can be specified into midbrain-like structures by Shh and FGF8 treatment. They demonstrated that simBOs derived from a PD patient with LRRK2-G2019S exhibited increased LRRK2 activity and a reduced DA neuronal number that can be mitigated by the treatment of these mutant organoids with the LRRK2 inhibitor PFE-360 [196]. Together, these recent findings demonstrate the versatility of using MOs in PD modeling that will pave the way for future PD therapeutics discoveries.

### 2.8. The Future of Organoid Research—State-Of-The-Art Assembloids, Organ-On-A-Chip, and Vascularization

Notwithstanding that the establishment of human brain organoids have provided us a means to gain insights into what went wrong in the brain of patients with PD and other neurodegenerative conditions, the majority of the brain organoids reported thus far represent specific brain regions in isolation that are often devoid of vascularization. This is a limitation as it is well established that progression of neurodegenerative diseases such as Parkinson’s involves a complex interaction between different cellular components and regions in the brain. Hence, the future of brain organoids research is to model multistystemic interactions crucial to our understanding of brain disease pathology. Assembloids, i.e., the fusion and functional integration of organoids of different cell types associated with different brain regions, represent the next step forward in disease modeling using the brain organoid system [244]. For a start, Chen et al. has recently reported the fusion of hMOs with human striatal organoids (hStrO) in an attempt to reconstruct the nigro-striatal pathway relevant to PD. They found that axons project from hMOs and target the fused StrO [245]. Further, given the attractive gut-brain axis proposal for the pathogenesis of PD [246], a gut-brain assembloid system may be used to interrogate the involvement of the gut in PD as well as to facilitate drug discovery. Notably, intestinal organoids are currently already in use for personalized medicine studies in the area of Cystic Fibrosis [247]. A parallel and exciting development is the integration of organoids via the Organ-on-a-Chip (OoC) technology. OoCs are systems of miniature tissues grown in microfludic chips designed to mimic human physiology with specific microenvironments [248]. To this end, several groups have modeled the gut-brain axis using 2D culture systems in OoCs; the simplest of which is the study of the blood–brain barrier using gut-epithelial and brain-endothelial cells [249]. Further, under the European Research Council-funded ‘MINERVA’ project, investigators aim to develop a multiorgan-on-a-chip platform to evaluate the impact of intestinal microflora on neurodegeneration. This bioengineering feat will integrate five organ-on-a-chip systems that hosts the gut microbiota, the gut epithelium, the immune system, the blood–brain barrier, and the brain. The brain chip is comprised of cultured neurons, astrocytes, and microglia embedded in an hydrogel matrix to obtain a 3D model [250]. As a refinement, Trapecar et al. [251] recently described the development of a mesofluidic platform technology to study gut-liver-cerebral interactions in PD that are bathed in a systematically circulated culture medium containing immune-related CD4+ regulatory T and T helper 17 cells. This approach results in an enhancement of features in the cerebral model on the chip that reflects the in vivo situation, and the finding that microbiome-associated short-chain fatty acids promote the pathological pathways of PD [251]. Notwithstanding the progress in the OoCs technology thus far, a limitation is that current systems utilizes mainly 2D cultures in the OoC which may not accurately capture what happens in vivo. Moving forward, it would be useful to tap on the OoCs technology to link the gut compartment to brain assembloids (or interconnected “multi-brain” chip) and investigate the propagation of α-Syn from the gut to the brain and the predilection of specific brain regions for α-Syn transmission.

A major problem that remains a challenge for the brain organoid model is the lack of vascularization, which not only restricts the growth of organoids in vitro but also promotes the onset of necrosis in the core region of the organoids where oxygenation is poor. Notably, co-culturing of endothelial cells and lung organoids in a microfluidic device apparently promotes angiogenesis-based perfusable vascular network into the core of the organoids that enable their survival for a longer term [252]. It remains to be seen if this technology is transferrable to brain organoids. Another recent approach in tackling the necrotic core issue is the use of spider silk microfibers with laminin to assemble a hierarchical 3D scaffold for iPSCs to self-arrange into MOs. The silk scaffolding forms microcavity networks that increase cell surface to volume ratio that facilitates oxygen flow and metabolic waste removal that result in a reduction in necrosis [253]. Similarly, others have used carbon fibers as a novel scaffold for MO generation to promote neuronal survival [199]. Beside these approaches, microfluidics [254] and milli-fluidics systems have been used as alternatives to orbital shaking to reduce the shear force and increase the oxygen and nutrient supply to organoid systems. Consequently, dopaminergic differentiation levels increased and necrotic cores reduced in size [190]. Interestingly, co-culturing with microglia also enhanced neuronal viability in organoids [213].

### 2.9. Use of mDA Precursor Cells for Neural Transplantation

Besides disease modeling and drug discovery, the iPSC technology also provides unique opportunities for regenerative medicine for human diseases, including PD. The use of PSC-derived DA precursors for PD transplantation was first described by Kriks et al., who demonstrated the ability of iPSC-derived mDA neurons to engraft successfully into the rodent models of PD induced by 6-hydroxy-dopamine [87]. Several similar studies have followed that collectively worked towards optimizing the transplantation efficiency and outcomes. To enhance the differentiation of hPSCs into transplantable neuronal progenitors and boost the engraftment efficiency, several groups have suggested the use of specific combinations of dopaminergic patterning factors. For example, Kirkeby and colleagues reported in 2017 that seeding human embryonic stem cell (hESC) onto Lam-111 and introducing FGF8b specifically from D9 to D16 of differentiation can yield more than 3.8 × 10^8^ progenitor cells from 1 × 10^6^ hESCs [255]. Interestingly, they found that FOXA2, LMX1A, and CORIN levels in the grafts may not positively correlate with the number of DA neurons. Rather, the markers EN1, SPRY1, WNT1, ETV5, and CNPY1, which are expressed by midbrain cells near the MHB, are associated with successful engraftment [255]. This may explain why older protocols produced markedly lower engraftment of TH+ cells post-transplantation (ranging from 6 to 54% between 1 to 4.5 months) despite 80% of cells being FOXA2 and LMX1A positive [87,148,256]. Another important component is GSK3 Inhibitor CHIR99021, which (as mentioned above) activates Wnt signaling to initiate mDA biogenesis. A bi-phasic Wnt activation is apparently necessary to enhance mDA neuron differentiation [174]. This requires an initial concentration of 0.7 μM CHIR99021 used in Knockout Serum Replacement (KSR)-free media that is increased to 7.5 μM within a narrow differentiation window. This CHIR99021 treatment paradigm significantly increases En1 levels, avoids aberrant patterning to diencephalic or hindbrain fates, and steers away from the unwanted generation of non-neural contaminants [174]. Notwithstanding the finding by Kirkeby et al. in 2017 that CORIN levels are poorly correlated with DA neuronal number in grafts, several groups continue to use CORIN as a marker to sort out DA progenitor cells generated from iPSCs, which seem to result in functional recovery in animals post-transplantation [256]. Conversely, another study suggested that transmembrane targets Alcam (also mentioned in [256]), Chl1, Gfra1, and Igsf8 that are highly expressed in mDA progenitors may be more sensitive targets as compared to CORIN for DA cell sorting they have improved selection efficiency for mDA progenitors [257]. Yet another candidate is Dlk1, which was reported in two studies as a marker of donor cells that was predictive of a positive post-transplantation outcome [150,255].

To further enhance the functional replacement of DA neurons in the PD brain, there is a need to refine current differentiation protocols towards increasing the population of A9-specific DA neurons in the grafts [258]. Nonetheless, the ability of PSC-based transplanted grafts to survive and restore motor deficits in mice, rats, or primates [87,150,255,259] in a manner that is comparable to the outcomes seen from the use of human fetal VM cells [259], is an incredible accomplishment on its own that has paved the way for their use as an attractive cell source for neural transplantation in PD patients. It is noteworthy to highlight that recently, a series of pioneering studies have also utilized cerebral organoids for engraftment in the brain of both mice [260,261,262] and cynomolgus monkey models [263]. Interestingly, these organoid transplants were well integrated into the host network with projections into the cortex, corpus callosum, and striatum [263]. Although these were done using cerebral organoids, it provides confidence that engraftment of midbrain organoids as a therapy for PD might be possible. In these instances, the host’s vascular network supported engrafted organoids with a steady blood supply, which may explain why these organoids engrafted well. This in turn may help circumvent the challenge of necrosis seen in organoids after prolonged culture in vitro [264], although it would mean that brain organoids would need to be grown in an in vivo environment.

### 2.10. From Bench to Bedside: Ongoing Human Trials Involving hPSC-Derived DA Precursors

Following the success in animal studies, a series of clinical trials involving hPSC-derived DA precursor cells for neural transplantation in PD patients have ensued. Before hiPSC-derived cells could be used for human trials, various stringent tests must be performed to check the product for morphology alterations, cell identity, viability, DNA fingerprinting, karyotype abnormalities, plasmid survival, adventitious agents, sterility, and endotoxins at several time points to ensure safety and quality control [173,265,266]. Thereafter, differentiated cells may undergo fluorescence-activated cell sorting (FACS) for markers of DA progenitors such as FOXA2+, TUJ1+, and CORIN+, to exclude any residual undifferentiated iPSCs. Additionally, some groups also checked for the positive expression of various markers of specific cell populations post-CORIN+ sorting by means of flow cytometry or quantitative reverse transcription polymerase chain reaction (RT-qPCR). These markers include OCT3/4, LIN28, and TRA-2-49 (for undifferentiated iPSCs), SOX1, PAX6, and KI67 (for proliferating early neural stem cells—NSCs), FOXA2, TUJ1, and LMX1A (for DA progenitors), and NURR1 and TH (for mature DA neurons). At this point, most cells should express DA progenitor markers, although a smaller population may also express mature DA neuronal markers [265]. In a related protocol developed by Schweitzer et al. [166], PSCs are expected to be positive for markers such Nanog, OCT4, and SSEA4 at day 0 of differentiation. By day 12, a 500-fold increase in FOXA2 and LMX1A should be expected. According to their study, two days prior to releasing DA progenitors for injection, at least 55% of cells must be positive for FOXA2 and LMX1A, and at least 10% of cells must be TH+, whereas OCT4, SSEA4, TPH2, 5-HT, DBH2 cannot be positive in more than 1% of the cells. On the day of final product release, cells were checked to ensure a viability above 70%, as well as a final confirmation that the sample is pathogen free [166]. Prior to implantation, further tests may be performed on the neurons to ensure that the kinetics of dopamine secretion and electrophysiological properties are similar to that of the native DA neurons in the SNpc [166,265]. Both iPSCs and DA progenitors may also be screened for genomic or epigenomic abnormalities using methods such as whole genome sequencing and whole exome sequencing. Epigenetic mutations may be checked by comparing the methylation ratio at the transcriptional start sites of cancer-related genes [265]. Further screening may also be done to detect for any neurodegeneration-related mutations [166], or for cancer-related mutations. For the latter, useful information may be found from the catalogue of somatic mutations in cancer (COSMIC) [266], Shibata’s gene list by the Pharmaceuticals and Medical Devices Agency of Japan (PDMA), and the human gene mutation database (HGMD) [265]. In addition to this, it would be useful to perform tumorigenicity, teratoma, toxicity, and biodistribution studies in immunodeficient (NOG/NSG) animals [173,265,266,267]. Teratoma assays may involve using spike controls via undifferentiated iPSCs to determine a threshold for the amount of iPSCs allowed in the final product [173,265], with the animals being observed for as long as possible to ensure that tumors or teratomas stemming from the injected cells do not develop further down the line [265]. For biodistribution studies, the goal is for samples to have undetectable levels of target cells in regions outside the central nervous system or in other organs [265,268]. In one case, sporadic and very low positive signals of human specific Alu repeat sequences was detected in other tissues and organs when injected into rodents, with the levels just above the limit of quantification. However, the low human DNA levels did not increase over 180 days and was attributed to suboptimal cell injection where accidental cell leakage may have occurred from the injection needle into the spinal fluid. [173]. A comprehensive list of quality assurance tests have been provided by others for readers who may be interested to learn more [173,174].

Meanwhile, several human trials involving iPSC-derived DA precursor cells have started across the world. In 2020, Schweitzer and colleagues [166] delivered patient specific-iPSC-derived DA progenitor cells bilaterally into the putamen of a 69-year-old man with a ten year history of progressive idiopathic PD. The patient subsequently showed an improvement in motor function, did not experience any adverse reactions 24 months post-surgery, and recorded a slight increase in ^18^F-DOPA uptake close to the graft sites [166]. In terms of allogeneic cell transplantations, the Center for iPS Cell Research and Application at Kyoto University (CiRA) transplanted iPSC-derived DA progenitors in seven PD patients without any apparent safety issues (clinical trial ID: UMIN000033564) [269]. The CiRA trial is part of GForce-PD [270], which is a global consortium consisting of teams from Japan, Europe, and the United States to conduct in-human clinical trials using hPSCs as a therapeutic source for PD. In addition, an ongoing trial involving 12 participants in North America is underway with an estimated completion date of May 2024 (clinical trial ID: NCT04802733) [271]. This trial uses hESCs that are differentiated into DA progenitors to be injected into the putamen of patients. Besides these, two other trials are currently in progress that use neural stem cells instead of DA progenitors for PD cell therapy. One is a Phase I/II, Open-Label Study conducted in China [272], while the other is a Phase 1 trial (clinical trial ID: NCT02452723) conducted in Australia that involves twelve individuals with PD. In the latter case, patients reported improvements over two years post-transplantation, both in terms of % OFF-time and PDQ-39 index [273]. While we await the outcomes from these studies, it is evident that the iPSC technology holds tremendous promise to revolutionize regenerative medicine for PD. Nonetheless, perhaps the next step is to seek a source of hypoimmunogenic iPSCs that are compatible with people with different human leukocyte antigen (HLA) types so that transplant recipients no longer need to be reliant on immunosuppressants.

## 3. Concluding Remarks

The advent of iPSC technology has provided us with a brand-new tool to model PD and expanded our armamentarium of experimental models to test and validate new drugs for this debilitating disease. Further, we anticipate exciting developments in organoid technology that hold promise to generate 3D brain structures that would mirror not just the human brain systems more faithfully, but also shed unique insights into brain-body interactions. At the same time, regenerative medicine for PD using iPSC-derived DA cells is moving at a breakneck speed, which fuels optimism regarding its routine application in the clinic for PD patients in the foreseeable future. At the end of the day, we would agree that it is through the fundamental understanding of the developmental process of mDA neurons that has allowed for such great strides in PD modeling to be achieved.

## Figures and Tables

**Figure 1 ijms-24-02523-f001:**
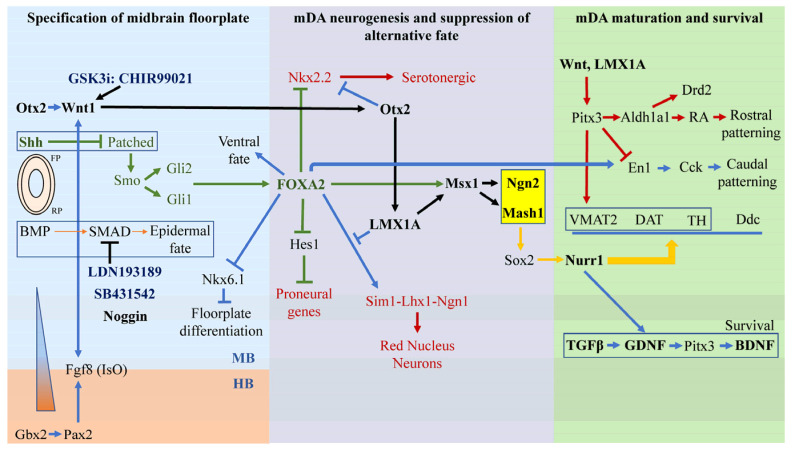
A plethora of small signaling molecules involved in 3 stages of mDA neurodevelopmental fate. The 3 stages are: 1. specification of midbrain floorplate; 2. specification of midbrain DA (mDA) neurons/mDA neurogenesis and suppression of alternative fate; and 3. mDA maturation and survival. The key pathways for specification of ventral midbrain fate are the Wnt1-Otx2-LMX1A-Msx1 pathway (black) and the Shh-FoxA2 pathway (green) that converged to Mash1 and Ngn2 (orange). Small molecules used in in vitro cultures mainly target the inhibition of SMAD signaling that drives epidermal fate and the activation of Wnt and Shh signaling.

**Figure 2 ijms-24-02523-f002:**
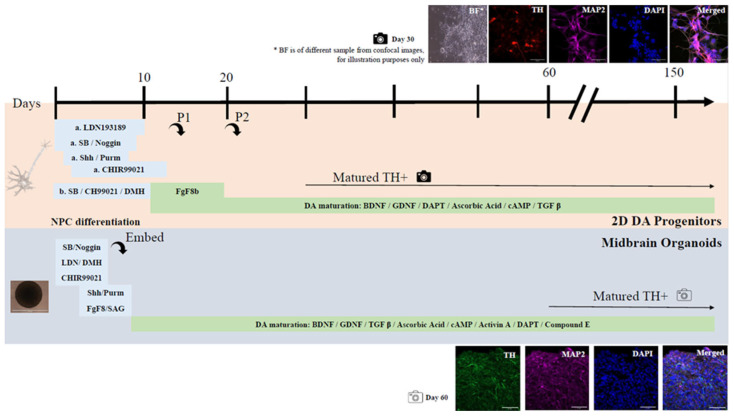
Comparison of 2D dopaminergic (DA) neuron and midbrain organoid (MO) derivation from iPSCs. (**top**): Typical timeline of 2D DA neuronal derivation adapted from various protocols (duration of each factor varies across protocols, approximate given). *Combo A:* LDN193189: ~Days 0–10; SB431542: ~Days 0–4; Noggin: ~Days 0–4; Shh: ~Days 1–6; Purmorphamine: ~Days 1–6; CHIR99021: ~Days 3–12; *Combo B:* SB431542/Dorsomorphin (DMH)/CHIR99021: ~Days 5–10; *DA maturation:* FgF8: ~Days 10–20; BDNF/GDNF/DAPT/Ascorbic acid/cAMP/TGFβ: ~Days 11 onwards. TH+ DA neurons mature by day 30 as shown in both brightfield (BF) and immunofluorescence staining. Note: BF and confocal images are from different samples, meant for illustration purposes. (**bottom**): Typical timeline of VMO derivation adapted from various protocols (duration of each factor varies across protocols, approximate given). SB431542: ~Days 0–6; Noggin: ~Days 0–6; CHIR99021: ~Days 0–6; Shh: ~Days 4–7; FgF8: ~Days 4–7; BDNF/GDNF/Ascorbic acid/cAMP: ~Days 8 onwards. TH+ DA neurons mature by day 60 as shown in immunofluorescence staining. Images are from our lab that we have generated from an adaption of Krik et al’s [87] (**top**) and Jo et al’s [95] protocols (**bottom**). The scale bar corresponds to 50 μM.

## Data Availability

Data is contained within the article or Appendix A.

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
