# Peer review of "From 2D to 3D: Development of Monolayer Dopaminergic Neuronal and Midbrain Organoid Cultures for Parkinson’s Disease Modeling and Regenerative Therapy"

_ijms, 2023, doi:10.3390/ijms24032523_

Round 1

Reviewer 1 Report

The manuscript entitled “From 2D to 3D: Development of Monolayer Dopaminergic Neuronal and Midbrain Organoid Cultures for Parkinson’s Disease Modeling and Regenerative Therapy” by Yeap et al summarizes the studies on 2D culture and 3D midbrain organoid (MO) models of Parkinson disease. In this manuscript authors reviewed different approaches for dopaminergic (DA) neuronal differentiation and abridged the key driving factors for DA neuronal differentiation, which could be utilized in development of iPSC-based future models of PD. Although this review has importance in its niche research area, authors are requested to address following comments:

Major Comments

1.     Are there specific reasons for selecting Kriks et al protocol for 2D culture and Jo et al method for 3D midbrain organoid culture in Figure 2? Please explain.

2.     It would be helpful if authors provide comparative summary of different protocols for midbrain 2D culture and 3D organoid culture in a Table.

3.     Page 9, paragraph 381. Authors stated “hMOs can be derived from a variety of PSCs [117] including embryonic stem cells 391 (ESCs) [112, 116] and iPSCs [111], or from neural progenitor cells (NPCs) [114, 115, 118, 392 119]”. Further they described several studies on NPCs derived hMO. Authors should also discuss studies on ESCs and iPSCs derived midbrain organoid culture including advantages/disadvantages of these models.

4.     Page 9, under paragraph “Using MOs in PD disease modeling and drug discovery”, authors should describe MO model of genetic PD and idiopathic PD under two separate sub-headings. Also, they should provide a comparative list of studies on genetic and idiopathic model of PD in an additional Table.

Minor comments:

1.     Authors should follow uniform reference style in the text. Page 6, Line 262 “Indeed, an animal study by Masoud et al. (2015) demonstrated that 262 DAT…..”. The reference year needs to be removed.

2.     Please add reference number to following lines:

Page 9, Line 382 and Line 387

Page 13, Line 621

Page 14, Line 655

Reviewer 2 Report

Yeap et al. have presented an excellent review on latest advancement in the development of human iPSC derived dopaminergic neuronal cultures and brain organoids, with emphasis on their application in disease modelling and potential clinical implication for Parkinson’s disease (PD). This review is well-written, informative, and comprehensive which is supported by sufficient and appropriate references from the recent literature. I agree with the authors that this review not only summarize the latest work done in the field but provides background information and a useful guide for novices entering the field who are interested to develop iPSC-based models of PD. I would recommend acceptance of this review for publication after addressing a few minor concerns as below.

1. This review covers a vast amount of technical information in different aspects of the topic which may not be easy to follow. It would be helpful to include summary tables in relevant sections (with citations) for easier reading and better understanding, especially for junior researchers. For example:

a.      Section 3: mDA neuronal development is a complicated process involving spatially and timely signalling of different factors and pathways. The authors have done a good job to summarise most of the relevant factors key to these processes. As a guide for inexperienced researchers in the field, it would be helpful to construct a simple table listing various key factors and their roles with references, in addition to the pathway diagram provided in Fig. 1.

b.      Section 4: A summary table listing uses of different protein factors in different developmental stages will be more practical and helpful.

2. Page 6, line 266: Please specify the relevant models used in the study mentioned.

3. Section 6 (page 10, line 436): The authors mentioned a study that hMO from patients carrying LRRK2 G2019S mutation may have shown a compensatory response to counteract the defective specification of mDA neurons induced by the mutation. Is it possible to elaborate and provide some comments whether such compensatory responses can happen in offspring of patients in real life? And how such experimental observation should be translated in the context of clinical situation. This may be a difficult question to address but it concerns the practicality of hMO in developmental research.

4. Section 6: A summary table of different PD MOs for different research aims is recommended.

5.  The authors can consider putting the discussion of current challenges and limitations of the technology as a separate section before the concluding remarks (i.e. Section 10).  

Thanks for a good review.
